# Propagation Losses Estimation in a Cationic-Network-Based Hydrogel Waveguide

**DOI:** 10.3390/mi13122253

**Published:** 2022-12-18

**Authors:** Carolina Pons, Josué M. Galindo, Juan C. Martín, Iván Torres-Moya, Sonia Merino, M. Antonia Herrero, Ester Vázquez, Pilar Prieto, Juan A. Vallés

**Affiliations:** 1Departamento de Física Aplicada-I3A, Facultad de Ciencias, Universidad de Zaragoza, C/P. Cerbuna 12, 50009 Zaragoza, Spain; 2Departamento de Química Inorgánica, Orgánica y Bioquímica, Facultad de Ciencias y Tecnologías Químicas-IRICA, Universidad de Castilla-La Mancha, 13071 Ciudad Real, Spain; 3Instituto Regional de Investigación Científica Aplicada (IRICA), UCLM, 13071 Ciudad Real, Spain

**Keywords:** hydrogel waveguides, attenuation coefficients, photographic method

## Abstract

A method based on the photographic recording of the power distribution laterally diffused by cationic-network (CN) hydrogel waveguides is first checked against the well-established cut-back method and then used to determine the different contributions to optical power attenuation along the hydrogel-based waveguide. Absorption and scattering loss coefficients are determined for 450 nm, 532 nm and 633 nm excitation. The excellent optical loss values obtained (0.32–1.95 dB/cm), similar to others previously described, indicate their potential application as waveguides in different fields, including soft robotic and light-based therapies.

## 1. Introduction

Optogenetics and photonic technologies are changing the near future of medicine. This new technology promotes the progress of early detection and diagnosis of diseases, as well as the possibility of a better understanding of biological systems [1,2]. In order to implement these light-based therapies, patient-friendly devices that can deliver light inside the body while offering tunable properties and compatibility with soft tissues are needed. The sensitivity and selectivity of the optical responses are strongly dependent on the characteristics of elements of optical devices, such as an optical waveguide. In this context, soft polymeric biomaterials have been used as alternatives to silica for fabricating optical waveguides [3]. Among these materials, optical waveguides based on hydrogels are arousing great interest, as they allow the distribution of light several centimeters deep in human tissues, without causing damage to the patient [4,5]. Hydrogels show excellent properties such as flexibility and biocompatibility. Furthermore, the ability of hydrogels to capture and release water converts them into fascinating materials for possible applications in soft robotics [6]. This behavior allows them to be employed as sensors [7], actuators [8] and bioinspired robots [9].

In this context; the optical and mechanical properties of the hydrogels can be fine controlled by adjusting the ratio between the monomers and the chain length of the crosslinking agent [10]. Moreover, the refractive index and thus its waveguide behavior can be modulated depending on the water content. Finally, variations in pore size, temperature, pH, ion concentration, degree of crosslinking, ligand attachment and interaction with other molecules give them interesting properties.

A key parameter to determine its capability to conduct light efficiently and its applicability is the attenuation experienced by optical power along the waveguide. The well-established cut-off method [11], in which the output power is measured for progressively shorter waveguide lengths, has a number of drawbacks: it is destructive, it requires a long-enough sample, high-quality waveguide edges and a certain time to prepare the measurement after each and every cut. One of the aims of this work is to explore the possibility of measuring the attenuation in hydrogel waveguides through a photographic record of the power distribution laterally diffused by the inhomogeneities of the material. With this method, referred to from now on as the photographic one, there is no need for cutting and realigning the setup, thus eliminating a significant source of measurement uncertainty and, also, employing a non-destructive procedure [12].

In the first part of the paper, the details of the photographic method are provided. The sample features are then presented and a comparison between attenuation measurements obtained by both methods is conducted, in order to check the reliability of the alternative procedure. Finally, attenuation measurements at different wavelengths are shown which, together with the CN–hydrogel absorbance spectrum, allows one to uncouple the different contributions to the propagation losses.

## 2. The Photographic Method

When light propagates along a guiding structure, optical power can be attenuated due to various phenomena (input/output coupling loss, propagation loss and bend radiation loss) [13]. The characterization of optical power attenuation establishes the material’s suitability for a given waveguiding functionality. 

A simple method to directly measure propagation losses is the so-called cut-back method, where the transmittances of waveguides having different lengths are compared. Since it is not easy to guarantee equal quality in different-length samples, measurements are usually performed by cutting a waveguide to change its length. This method has the advantage of its simplicity, but it is a destructive technique and requires repetitive high-quality waveguide edges so that equal input/output coupling efficiencies are achieved for every measurement.

An alternative waveguide-loss characterization method (the photographic method) that overcomes the main drawback of the cut-back method is the measurement of the scattered-light intensity distribution along the waveguide. Provided the waveguide is nearly uniform, this intensity is proportional to the guided-light intensity at each point. The scattered light streak can be registered by a camera and the attenuation coefficients are obtained through image processing. This way, propagation loss coefficients can be determined for non-negligible-scattering waveguides with nondestructive and noncontact measurements with a relatively simple configuration. 

A scheme of the experimental set-up for the implementation of the photographic method is shown in Figure 1a. At the moment of the measurement, each cylinder-shape hydrogel sample is kept fixed and straight by placing it on the groove of a suitable piece (not shown in the scheme). Light from a laser is then end-coupled and images of the illuminated sample are taken by means of a computer-controlled Canon EOS 1000D camera, allowing an independent detection in the R, G and B channels. For each sample, pictures are taken seeking maximum contrast without reaching saturation. In order to achieve this, apart from the camera parameters such as exposure time, ISO and aperture, the input laser power was also modified, within the range of a few mW. Figure 1b shows an example of the pictures obtained with λ = 450 nm excitation. In order to extract the longitudinal optical intensity distribution from each picture, several operations have to be performed. First, an area of the picture is selected, avoiding the section close to the laser power input, where coupling effects may give rise to an undesired contribution to a lateral light and, in general, both saturated and very dark regions are also discarded. In addition, it is convenient to avoid brilliant points corresponding to damaged spots caused by sample manipulation. The area selected determines a set of M × N pixels, with M and N being the number of pixels selected in the X and Y directions, respectively (see Figure 1b). We pay attention to the information they contain at the channel corresponding to the laser wavelength employed. This information is a number between 0 and 255 not proportional to but directly correlated with the light energy received by the pixel. By application of a transform inverse to the one shown in [14], we obtain a matrix of N × M values proportional to the relative exposure received at each pixel of the selected region. Finally, in order to reduce noise caused by surface imperfections, the N values corresponding to each matrix column are averaged so that an array of M values is obtained. There was the option of directly selecting an M × 1 array of pixels, but we have checked that considering an M × N matrix and performing a transversal average is worthwhile, as it provides a significant noise reduction. 

The correlation between pixel position at the image and coordinate at the object can be easily obtained by means of a picture of two points aligned with the X axis and whose distance is accurately measurable.

## 3. Experimental Results and Analysis

### 3.1. The Samples

The hydrogels used were based on a cationic electroactive network (CN), shown in Figure 2 and previously reported by some of the authors [15] (with applications in the field of soft robotics [16] and with self-healing ability [17]). Further details of the CN-hydrogel material and synthesis can be found in Appendix A. This kind of hydrogel is able to retain a certain and constant amount of water in its 3D polymer structure at ambient conditions (25 °C, 1 atm), reaching an “equilibrated” state and remaining stable. In addition, this hydrogel shows high flexibility together with excellent mechanical properties in the equilibrated state (E = 457 kPa) and potentially favorable optical properties, such as a high degree of transparency, which is crucial for low optical loss [1]. Thus, the measurements were performed in CN-hydrogels in the equilibrated state in order to have a reproducible system.

The hydrogel samples used for the measurements were cylinder-shaped, with diameters between 3 mm and 4 mm and lengths between 6 cm and 10 cm

### 3.2. Comparison between Photographic and Cut-Back Methods

In this section, we compare the results obtained on a similar sample using the photographic and cut-back methods. Figure 3 shows an example of the relative exposure obtained after processing an image obtained in the B channel of the scattered-light intensity distribution for λ = 450 nm excitation. There, results are fitted to an exponential function, P(x) = P_0_ exp(-αx), where P_0_ and α are the fitting parameters and, in particular, α represents the attenuation coefficient. The value obtained for the data in Figure 3 is α = 0.45 cm^−1^.

The cut-back method is now applied to a similar sample, with the signal measured at the waveguide output end plotted in Figure 4 vs. the waveguide length after successive cuts. The same λ = 450 nm feeding laser is used for the hydrogel-based waveguide excitation. A value α = 0.47 cm^−1^ is obtained for this coefficient by means of the cut-off method. Figure 4 shows the results obtained and the fitting decay.

The excellent agreement in the values of the attenuation coefficients obtained by both methods (less than a 4% disagreement) definitely supports the validity of the photographic methods for attenuation measurements in this kind of sample.

### 3.3. Determination of the Attenuation Coefficients

The measurement procedure in the photographic method avoids any influence from I/O coupling or bending losses; therefore, the attenuation coefficients are determined to account for the possible causes of propagation losses. In our experiments, propagation losses can be considered to be caused by material absorption and by optical power scattering (due to small-size material inhomogeneities and surface roughness) that can be also understood as mode conversion from guided to radiation modes [12]. Hence, the experimental decay equation can be factorized according to P(x) = P_0_ exp(-α_a_x) exp(-α_s_x), where the coefficients α_a_ and α_s_ account for absorption and scattering losses, respectively, and α = α_a_ + α_s_. 

Material absorbance was experimentally measured, with the absorbance spectrum of the CN-hydrogel shown in Figure 5.

From the measured absorbance values, the absorption loss coefficients can be calculated as: α_a_(450 nm) = 0.6587 cm^−1^, α_a_(532 nm) = 0.4377 cm^−1^ and α_a_(633 nm) = 0.3108 cm^−1^, although it seems reasonable to assume that a background value α_a0_ could be affecting these coefficients.

Since the surface imperfections effect was minimized by the transversal average conducted during image processing, the remaining scattering loss can be assumed to follow the Rayleigh λ^−4^ law and, accordingly, the ratios between the scattering loss coefficients are α_s_ (450 nm)/α_s_ (633 nm) = 3.9153 and α_s_ (532 nm)/α_s_ (633 nm) = 2.0043. In order to determine the wavelength-dependent α_s,_ we will take advantage of the independent detection in the R, G and B channels performed by the camera. A sample was excited using laser sources centered at 532 nm and 630 nm, with the relative exposure distributions along the waveguide obtained shown in Figure 6. The corresponding fittings provide α = 0.21 cm^−1^ (λ = 532 nm) and α = 0.07 cm^−1^ (λ = 633 nm).

A set of Equations (1)–(3) can be established by considering the relationship between attenuation, absorption and scattering loss coefficients for each wavelength:(0.3108 − α_a0_) + α_s_(633 nm) = 0.07354(1)
(0.4377 − α_a0_) + 2.0043α_s_(633 nm) = 0.20894(2)
(0.6587 − α_a0_) + 3.9153α_s_(633 nm) = 0.45067(3)

Then, using α_a0_ and α_s_(633 nm) as fitting parameters of a least squares procedure, the absorption and scattering loss values (in dB/cm) in Table 1 have been obtained.

Although scattering losses are certainly significant, absorption losses are clearly the largest contribution to propagation losses. Their strong wavelength dependence suggests a careful selection of the operating wavelength is necessary, where possible. However, taking into account the several-centimeters length of the hydrogel-based waveguides used for light distribution in human tissues, the values obtained for the loss coefficients confirm their suitability for this functionality. 

The outcomes obtained indicate that this hydrogel presents excellent optical loss values similar to others previously reported and collected in Table 2.

The waveguide properties found for this CN-based hydrogel, as well as their use in soft robotics [14] and self-repair [15] described by our group, increase their potential and their use in the field of soft robotics and perhaps in light-based therapies. It should be noted that the hydrogel also presents excellent transparency and high stability due to the fact that it is in a stabilized state and, therefore, there is no loss of water, thus maintaining its structure. In addition, the structure can be chemically modulated in order to modulate the properties.

## 4. Conclusions

A method to determine the contributions to the attenuation of optical power along a hydrogel-based waveguide has successfully been validated against the well-established but experimentally more demanding cut-back method. This methodology allows the determination of this parameter without destroying the sample. Using this method, from the hydrogel absorption spectrum and the lateral-scattering intensity longitudinal distributions in the three camera R, G and B channels, the absorption and scattering losses coefficients of the hydrogel-based waveguides can be readily determined. In particular, for the CN-based hydrogel waveguides used for the measurements, the results confirm their good optical loss values, similar to those previously described. These findings indicate their potential application as a biomaterial in soft robotics and light-based therapies. 

## Figures and Tables

**Figure 1 micromachines-13-02253-f001:**
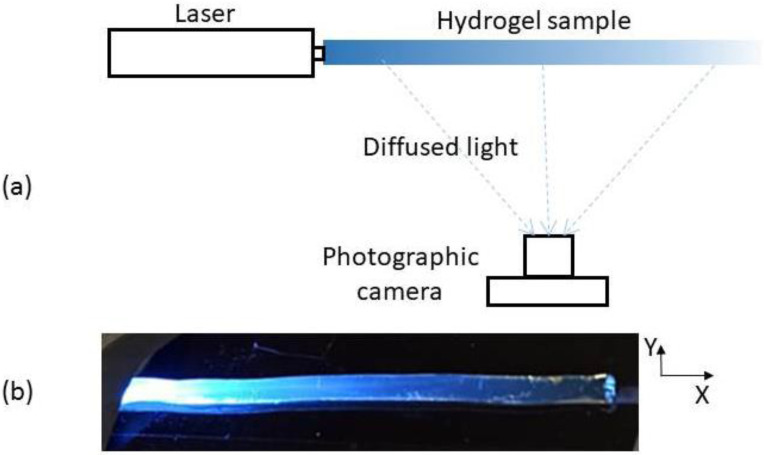
(**a**) Scheme of the experimental setup. Sizes and distances are not drawn to scale. (**b**) Picture of a hydrogel sample fed at its left end with the laser centered at 450 nm. The coordinate system to be employed later is specified.

**Figure 2 micromachines-13-02253-f002:**
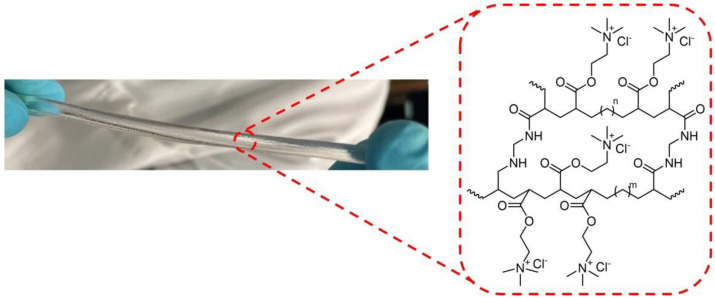
Chemical structure of the CN-hydrogel and digital image of the hydrogel cylinder used for the measurements.

**Figure 3 micromachines-13-02253-f003:**
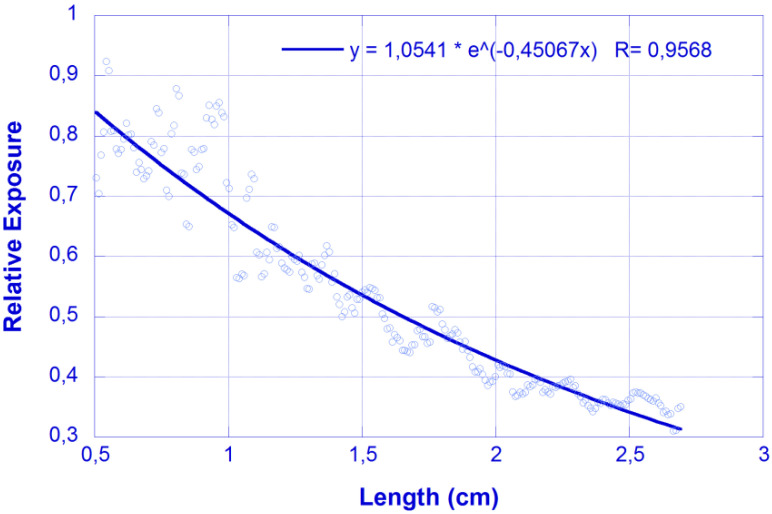
Relative exposure vs. longitudinal coordinate (photographic method). Feeding laser λ = 450 nm. Dots: experimental values; Solid line: fit to exponential decay.

**Figure 4 micromachines-13-02253-f004:**
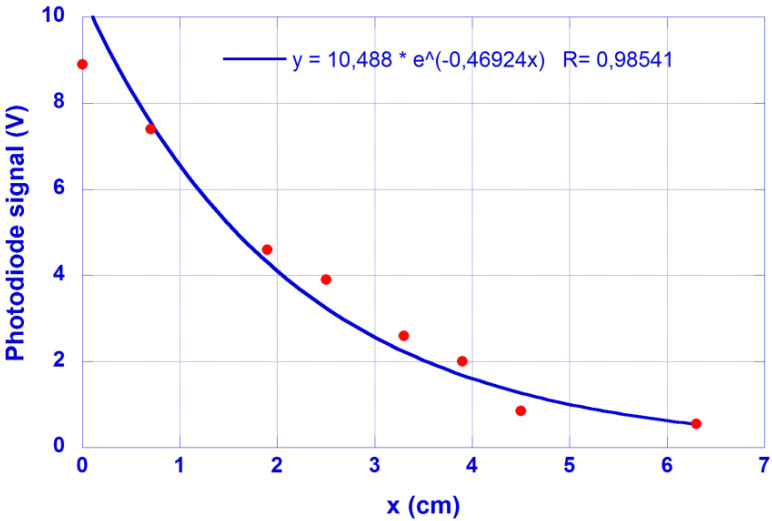
Photodiode signal at the sample output end as a function of its length (x + 4 cm). Feeding laser @ 450 nm. It is a similar sample as employed in the photographic method.

**Figure 5 micromachines-13-02253-f005:**
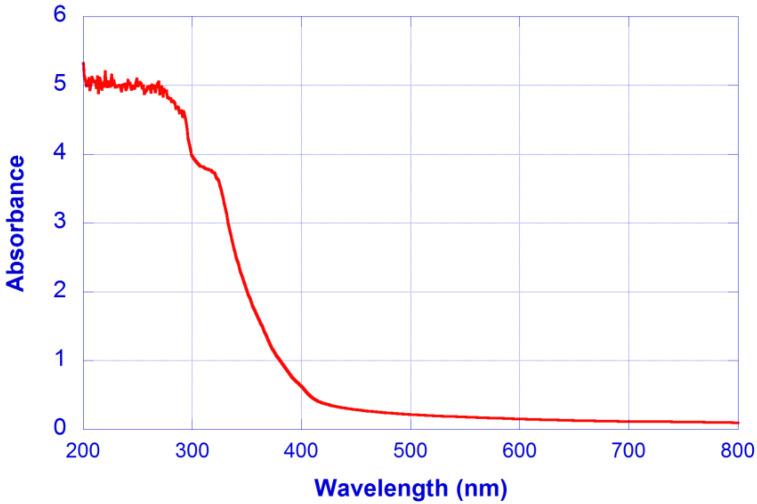
Absorbance spectrum of the CN-hydrogel.

**Figure 6 micromachines-13-02253-f006:**
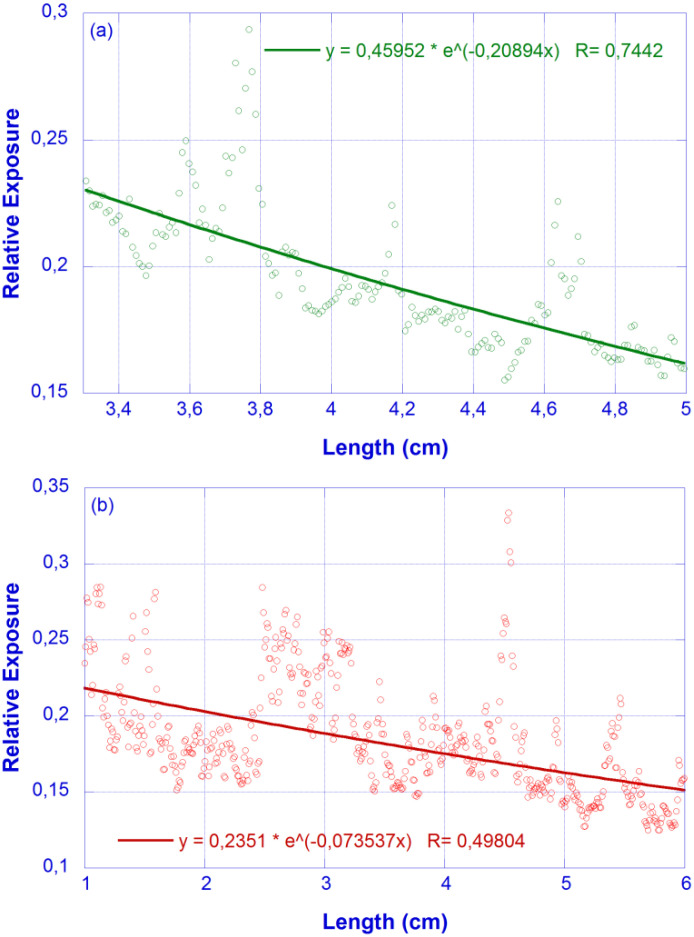
Relative exposure vs. X coordinate (photographic method). Feeding lasers: (**a**) λ = 532 nm (green, top); (**b**) λ = 632 nm (red, bottom). Dots: experimental values; Solid line: fit to an exponential decay.

**Table 1 micromachines-13-02253-t001:** Contributions to attenuation at the three exciting wavelengths used for the measurements.

Wavelength (nm)	Absorption Loss (dB/cm)	Scattering Loss (dB/cm)	Optical Loss (dB/cm) ^(a)^
450	1.78	1.73 × 10^−1^	1.95
532	8.24 × 10^−1^	8.79 × 10^−2^	0.91
633	2.73 × 10^−1^	4.43 × 10^−2^	0.32

(a) Optical loss = Absorption loss + Scattering loss.

**Table 2 micromachines-13-02253-t002:** Optical loss values of several hydrogels previously described in the literature.

Materials	Wavelength (nm)	Optical Loss (dB/cm)	References
PEGDA Hydrogels	400–800	1.25–5.15	[18]
CDs-PEGDA Hydrogels	405	0.55–1.1	[18]
Silicone-based Hydrogels	632	7.5	[19]
PEG-Hydrogels	450–550	0.17–0.68	[4]
PEGDA-DDT	405–520	0.1–0.4	[2]
PEGDA 700–alginate hydrogel	492	0.32–0.42	[20]
PAM–alginate hydrogel	400–700	0.4	[21]
PAM hydrogel	532	1–11	[5]

PEGDA: Poly(ethylene glycol) diacrylate. PEG: Poly(ethylene glycol); PAM: Polyacrylamide.

## Data Availability

Data are available on request.

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
