# Peer review of "Propagation Losses Estimation in a Cationic-Network-Based Hydrogel Waveguide"

_micromachines, 2022, doi:10.3390/mi13122253_

Round 1
Reviewer 1 Report
The authors present a method based on the photographic recording of the power distribution laterally diffused by cationic-network (CN) hydrogel waveguides. Unfortunately, I think the amount of data in the draft needs to be supplemented and the innovative description is insufficient. А major revision is required. For instance, the following comments should be addressed:
1. The abstract does not highlight the innovation of the manuscript and key performance indicators should be added.
2. A comparison table of performance indicators with those reported in the relevant literature should be added to highlight the strengths of the manuscript.
3. The parameter information of the waveguide is missing in the manuscript.
4. In Figure 4, the amount of data is not enough to support the fitting function, which needs to be supplemented.
5. The quality and scale of figures need to be adjusted, such as Figures 1 and 2.
Reviewer 2 Report
The paper demonstrates a nondestructive method for measuring the propagation loss in hydrogel waveguide. Lateral scattering of the waveguide is registered by a camera, and the attenuation coefficients are obtained by fitting the recorded intensity. The authors further analyze the contributions from absorption and scattering losses in RGB channels. Overall, the proposed method is clearly presented and the analyses are solid. However, I have some concerns that need to be addressed prior to any decision.
1. Regarding the measurement of waveguide loss through lateral imaging, similar work has been reported (Wang, Fengtao, et al. "Precision measurements for propagation properties of high-definition polymer waveguides by imaging of scattered light." Optical engineering 47.2 (2008): 024602.). The authors should cite this work and better clarify the core difference in relation to previous studies.
2. The measurement results from the photographic method, especially in Fig. 6, are significantly deviated from an exponential function. One may suspect that the obtained coefficients or the theoretical models are inaccurate. The authors should analyze the sources of the deviation.
3. In the introduction, the authors emphasize the destructive nature of cut-back method since hydrogel waveguide could be embedded in human body. However, the proposed photographic method cannot be directly applied to this scenario either since visible light cannot easily penetrate through human tissues. The authors are recommended to comment on that.
Round 2
Reviewer 2 Report
In this version, the authors address most of my concerns satisfactorily. Thus, I support acceptance of the paper.